# Pancreatic beta-cell IL-22 receptor deficiency induces age-dependent dysregulation of insulin biosynthesis and systemic glucose homeostasis

Haressh Sajiir [1,2], Kuan Yau Wong[1,2], Alexandra Müller[1,2], Sahar Keshvari [1,2], Lucy Burr [1,2,3], Elena Aiello[4], Teresa Mezza[5,6], Andrea Giaccari [5,7], Guido Sebastiani[4], Francesco Dotta [4,8], Grant A. Ramm [2,9], Graeme A. Macdonald [2,10], Michael A. McGuckin[11], Johannes B. Prins[12] & Sumaira Z. Hasnain [1,2,13] ✉

The IL-22RA1 receptor is highly expressed in the pancreas, and exogenous IL-22 has been shown to reduce endoplasmic reticulum and oxidative stress in human pancreatic islets and promote secretion of high-quality insulin from beta-cells. However, the endogenous role of IL-22RA1 signaling on these cells remains unclear. Here, we show that antibody neutralisation of IL-22RA1 in cultured human islets leads to impaired insulin quality and increased cellular stress. Through the generation of mice lacking *IL-22ra1* specifically on pancreatic alpha- or beta-cells, we demonstrate that ablation of murine beta-cell *IL-22ra1* leads to similar decreases in insulin secretion, quality and islet regeneration, whilst increasing islet cellular stress, inflammation and MHC II expression. These changes in insulin secretion led to impaired glucose tolerance, a finding more pronounced in female animals compared to males. Our findings attribute a regulatory role for endogenous pancreatic beta-cell IL-22ra1 in insulin secretion, islet regeneration, inflammation/cellular stress and appropriate systemic metabolic regulation.

Interleukin-22 (IL-22) is a member of the IL-10 family of cytokines and is known to be a potent suppressor of cellular stress[1,2]. This cytokine signals via binding to a transmembrane complex consisting of 2 different heterodimeric subunits; interleukin-22 receptor subunit alpha (IL-22RA1) and interleukin-10 receptor subunit beta (IL-10RB2)[3]. Whilst IL-10RB2 is constitutively expressed, IL-22RA1 expression is restricted to epithelial and secretory cells[4]. The pancreas is responsible for maintaining glucose homeostasis through the secretion of glucagon and insulin from pancreatic alpha- and beta-cells, respectively. Interestingly, data from publicly available RNA-seq datasets indicates that the IL-22RA1 receptor is most highly expressed in the pancreas[5,6]. Bulk RNA-seq data from a study by Asplund et al. also shows that *IL-22RA1*

*expression* is upregulated in the pancreatic islets of patients with type-2 diabetes (T2D)[7]. Due to the relatively large volume of proteins being secreted by pancreatic islets, these cells are particularly susceptible to cellular stress[8]. Endoplasmic Reticulum (ER) and oxidative stress play integral roles in beta-cell dysfunction, disrupting protein folding and decreasing insulin production[9–12]. Progressive beta-cell dysfunction is characterised by increased ER stress and altered insulin processing over time, leading to a greater proportion of total insulin being secreted in the immature form of proinsulin[13].

Several studies have alluded to IL-22 signaling playing a key role in maintaining appropriate metabolic function. Wang et al. showed that *IL-22ra1⁻/⁻* mice on a high-fat diet for 12 weeks developed more severe

**Fig. 1 | Endogenous IL-22RA1 signaling regulates oxidative homeostasis and insulin quality control in human islets. a** mRNA fold change of *IL-22RA1* gene expression in healthy lean versus T2D human donor islets, relative to control *GAPDH*. **b** Total insulin, **c** proinsulin secretion (ng/10 islets/30 min), and **d** proinsulin: insulin ratio from human donor islets during glucose stimulated insulin secretion following treatment 10 µg mL⁻¹ anti-IL22RA1 (*p = 0.0341*). **e** mRNA fold change of *spliced XBP-1* (*sXBP-1; p = 0.0225*) and **f** *NOS2* in human donor islets, relative to control *GAPDH* following treatment with 10 µg mL⁻¹ anti-IL22RA1. **g** Intracellular nitrite production (µM) in human donor islets following 24 h treatment with 10 µg mL⁻¹ anti-IL22RA1. All graphs presented as Mean ± SEM. **a** *n* = 3 biologically independent donors, Two-tailed *Mann-Whitney* Test; **b**–**g** *n* = 3 biologically independent human islet donors, Kruskal-Wallis test with Dunn's multiple comparisons test. **p* < 0.05; n.s., non-significant. *versus vehicle (anti-IgG) control. Source data are provided as a Source Data file.

glucose intolerance and insulin resistance than wild-type mice[14]. We have previously shown that exogenous administration of IL-22 reduced ER and oxidative stress in human pancreatic islets, thereby promoting the secretion of high-quality insulin[2]. Furthermore, human donor islets treated with an IL-22RA1 neutralising antibody also had increased nitrite and reactive oxygen species (ROS) production, as well as increased genetic markers of cellular stress[2]. This suggested that endogenous IL-22RA1 signaling in pancreatic islets promotes appropriate insulin secretion through the suppression of cellular stress. Moreover, activation of IL-22RA1 signaling (exogenous treatment of IL-22) leads to a decrease in inflammation[2], which we have recently shown is attributed to its ability to suppress MHC II on epithelial cells[15]. However, the role of endogenous IL-22/IL-22RA1 signaling in the endocrine pancreas is incompletely understood despite the receptor profile of IL-22RA1. Therefore, by generating mice that lack IL-22ra1 on pancreatic alpha- and beta-cells we highlight a novel role of IL-22RA1 beta-cell signaling in maintaining insulin biosynthesis and regulating glucose homeostasis.

Here, we show the absence of beta-cell IL-22RA1 signaling leads to significant impairments in insulin biosynthesis and systemic glucose homeostasis. Mice deficient in beta-cell IL-22RA1 exhibit increased islet cellular stress, inflammation, and MHC II, alongside a reduction in insulin quality and secretion efficiency. These changes culminate in pronounced glucose intolerance, a phenomenon more severe in female mice. Our data highlight the potential of targeting IL-22RA1 signaling pathways as therapeutic avenues for preserving pancreatic function and managing diabetes.

## Results

### IL-22RA1 signaling inhibits cellular stress signaling and maintains insulin production and secretion in pancreatic beta-cells

To ascertain the role of endogenous IL-22RA1 in the pancreas, we obtained human islets from candidates to pancreatectomy[16] with normal glucose tolerance; NGT), those with impaired glucose tolerance (IGT) and T2D. The expression of *IL-22RA1* was increased in human pancreatic islets from IGT and T2D patients, compared to NGT

(Fig. 1a, Supplementary Fig.1a). Gene expression of IL-10RB subcomponent followed a similar trend, whilst expression of the *IL-20RA* and *IL-20RB* remained unchanged (Supplementary Fig. 1b–d). This suggests that IL-22 signaling, and not IL-20 or IL-24 signaling, is associated with hyperglycaemia. To explore this, we cultured healthy human donor islets with anti-IL-22RA1, and measured glucose stimulated insulin secretion. Whilst total insulin secretion from anti-IL-22RA1 treated human islets remained unchanged, neutralising IL-22RA1 receptor signaling resulted in a disproportionate increase in proinsulin secretion. Importantly, levels of proinsulin induced by anti-IL-22RA1 stimulation were comparable to levels of proinsulin secreted from T2D islet donors (Fig. 1b–d). In T2D, peripheral tissue insulin resistance can lead to compensatory increased insulin demand, where pancreatic beta-cells increase insulin production to overcome resistance[17]. Increased beta-cell workload leads to an increase in circulating levels of proinsulin:insulin ratio in subjects with insulin resistance compared with controls, which is often associated with an increase in ER and oxidative stress[13,18,19]. This reflects the beta cells' strained capacity to process and convert proinsulin to insulin efficiently to cope with increased demand for insulin[18]. Consistent with the increase in proinsulin secretion from islets after, neutralisation of IL-22RA1 signaling, there was an increase in ER stress, as determined by *spliced-XBP-1* (*sXBP-1*) mRNA levels. This was accompanied by an increase in *nitric oxide synthase 2* (*NOS2*) mRNA, and intracellular nitrite protein production (Fig. 1e–g). This corroborates the potential of endogenous IL-22RA1-signalling in driving pathways that inhibit cellular oxidative and ER stress and improve insulin quality in pancreatic beta-cells[2].

Pancreatic beta-cells, the most abundant cell type in human pancreatic islets, produce, store, and secrete insulin. To explore the pathways downstream of IL-22ra1 signaling, we treated *MIN6N8* mouse insulinoma beta-cells with IL-22 and performed RNA sequencing. Differential gene expression analysis identified key genes involved in insulin secretion and processing, including *regulator of G protein signaling 4 (Rgs4), gamma-aminobutyric acid B receptor 2 (Gabbr2), proprotein convertase subtilisin/kexin type 1 inhibitor (Pcsk1n)* and *MAF BZIP transcription factor A (Mafa)*, to be highly upregulated with IL-22

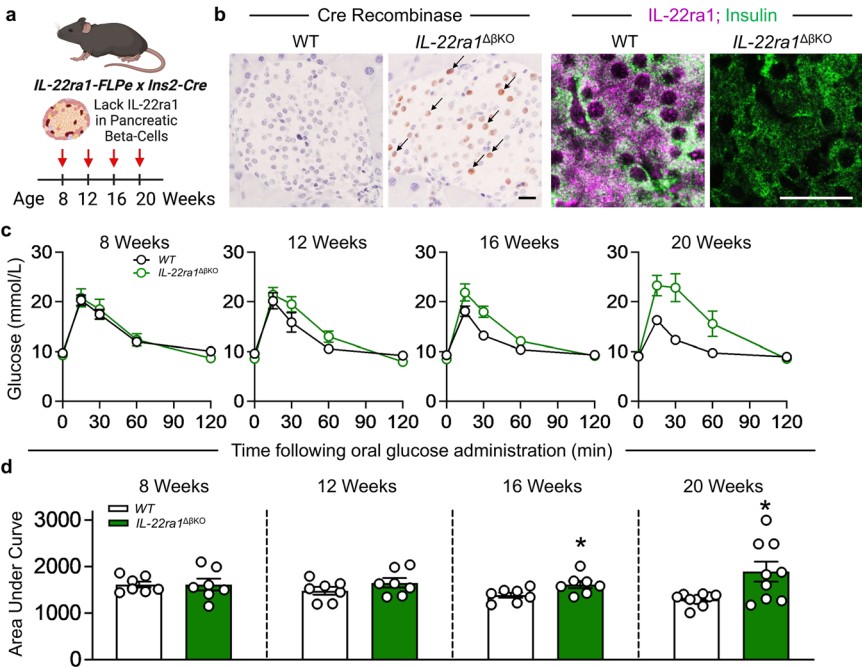

**Fig. 2 | Ablation of endogenous IL-22ra1 signaling in pancreatic β-cells leads to impaired glycemic tolerance with age. a** Experimental schematic, created with BioRender.com, released under a Creative Commons Attribution-NonCommercial-NoDerivs 4.0 International license. **b** Pancreatic sections of β-cell *IL-22ra1* knockout animals and their wildtype littermate counterparts stained for cre-recombinase, IL-22ra1 and Insulin. **c** Changes in glucose tolerance following oral glucose administration in animals with age. **d** Area under the curve during oral glucose tolerance tests in animals with age (16 weeks *p = 0.0380*; 20 weeks *p = 0.0146*). All graphs are presented as Mean ± SEM. Female animals; *n* = 7 biologically independent animals (8–16 weeks), 9 biologically independent animals (20 weeks), Two-tailed unpaired Student's t-test. *$p < 0.05$; n.s., non-significant. *versus wildtype (IL-22ra$^{fl/fl}$) littermates. Scale Bar: 20 um. Source data are provided as a Source Data file.

treatment (Supplementary Fig. 2a). Pathway analysis revealed that activation of IL-22ra1 in MIN6N8 cells upregulated multiple genes involved in insulin secretion signaling pathway and insulin-like growth factor I (IGF-1) signaling (Supplementary Fig. 2b). The activation of antioxidant pathways driven by NRF-2 were also confirmed to be upregulated in response to IL-22 (Supplementary Fig. 2b), as we have previously shown[2]. Importantly, IL-22 treatment downregulated expression of key genes involved in disease pathways including endocrine pancreatic dysfunction, Diabetes mellitus, severe pancreatic disorders, and impaired glucose tolerance (Supplementary Fig. 2c). Collectively, these data suggest that IL-22ra1 signaling helps to maintain beta-cell homeostasis and insulin secretion.

### IL-22ra1 deletion in pancreatic beta-cells causes hyperglycemia and glucose intolerance

To explore the endogenous role of IL-22ra1-signaling, we generated mice lacking the *IL-22ra1* receptor specifically on pancreatic beta-cells by crossing *Ins2-Cre* mice with *IL-22ra1-FLPe* animals; herein referred to as *IL-22ra1*$^{ΔβKO}$ mice (Fig. 2a). Cre-recombinase staining confirmed specificity to the mouse pancreata and the absence of IL-22ra1 in the beta-cells was confirmed by co-staining with insulin (Fig. 2b). Females *IL-22ra1*$^{ΔβKO}$ animals exhibited onset of severe glucose intolerance beginning at 16 weeks of age (Fig. 2c, d). In contrast, male animals did not show a statistically significant change at this age. By 20 weeks, although not reaching statistical significance, there was a notable trend indicating that male animals were also on the trajectory towards glucose intolerance (Supplementary Fig. 3a, b). Concurrently, at this age the area under the curve (AUC glucose) for the oral glucose tolerance test (OGTT) was 50% higher in *IL-22ra1*$^{ΔβKO}$ females than in wildtype littermate controls (Fig. 2d). This could be explained by *IL-22ra1* being expressed more highly in females compared to males[20]. Impaired glycemic control is often characterized by a marked increase in peripheral insulin resistance[21,22]. However, we observed no significant difference in insulin sensitivity in the *IL-22ra1*$^{ΔβKO}$ animals

(Supplementary Fig. 4a, b). IL-22ra1 is also known to be highly expressed on pancreatic alpha-cells. To rule out the potential contribution of α-cells, we generated *IL-22ra1*$^{ΔαKO}$ mice (Gcg-Cre x *IL-22ra1-FLPe*), which lack the *IL-22ra1* receptor specifically on pancreatic alpha-cells (Supplementary Fig. 5a). Ablation of α-cell-IL-22ra1 signaling had no effect on body or pancreas weights, serum glucagon, glucose tolerance, insulin sensitivity or overt endocrine/exocrine pathology (Supplementary Fig. 5b–g); confirming the metabolic phenotype observed in *IL-22ra1*$^{ΔβKO}$ animals was specific to pancreatic β-cell IL-22ra1 signaling.

### Endogenous IL-22ra1 signaling in pancreatic beta-cells is a key regulator of insulin biosynthesis

Hyperglycemia and glucose intolerance in the absence of alterations in systemic insulin sensitivity strongly implied that there was a potential defect in insulin biosynthesis or secretion in *IL-22ra1*$^{ΔβKO}$ animals. Immunostaining for proinsulin and insulin revealed a slight increase in proinsulin storage in the *IL-22ra1*$^{ΔβKO}$ animals (Fig. 3a, b). However, *IL-22ra1*$^{ΔβKO}$ mice had significantly less postprandial serum total insulin and proinsulin, with no changes in proinsulin: insulin ratio (PI:I) (Fig. 3c, d). mRNA analyses of whole pancreata from these animals also revealed significant suppression of key genes involved in insulin signaling and secretion (Fig. 3e, f). We confirmed that these changes were only observed with insulin as glucagon secretion in *IL-22ra1*$^{ΔβKO}$ animals remained unaltered (Supplementary Fig. 4c).

To explore the effects of metabolic stress on insulin secretion, we challenged *IL-22ra1*$^{ΔβKO}$ mice and their littermate controls with a high-fat diet (HFD) for 12 weeks. Whilst no major changes were observed in insulin sensitivity, the HFD significantly aggravated glycemic control in the *IL-22ra1*$^{ΔβKO}$ animals compared to their littermates on a HFD, and age-matched *IL-22ra1*$^{ΔβKO}$ animals on a normal chow diet (Supplementary Fig. 6a, b). The ratio of serum proinsulin to total insulin is known to be elevated with a HFD, which indicates inefficient insulin maturation in pancreatic beta-cells[2,11]. Interestingly proinsulin: insulin ratios were

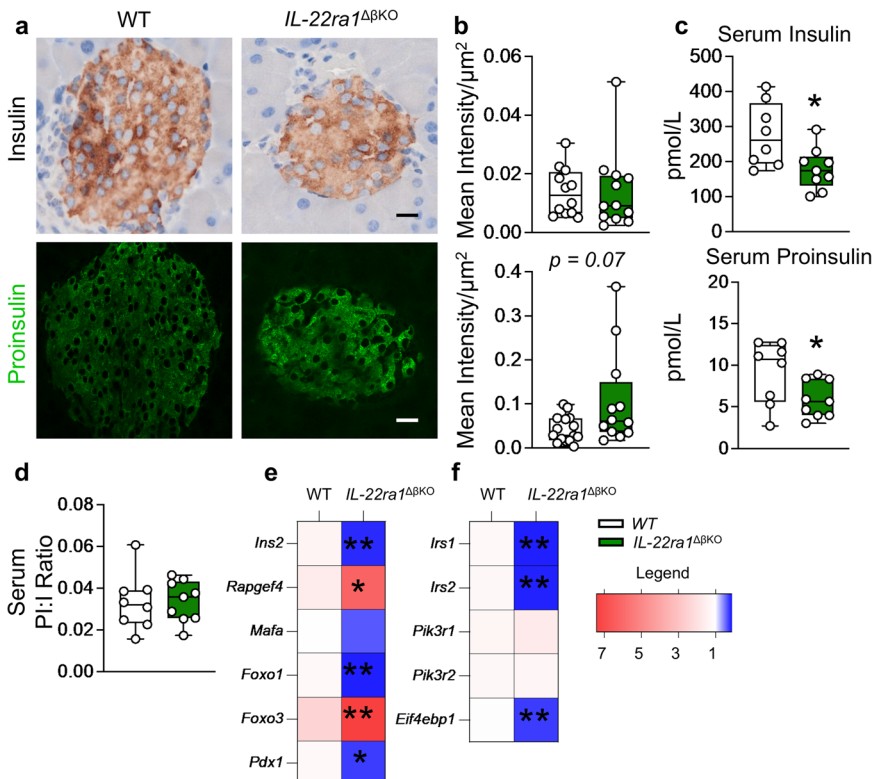

**Fig. 3 | Endogenous pancreatic β-cell IL-22ra1 signaling is a key regulator of insulin biosynthesis. a** Pancreatic sections stained for insulin and proinsulin. **b** Mean intensity per islet area of insulin and proinsulin. **c** Total serum insulin ($p = 0.0175$) and proinsulin ($p = 0.0452$) in animals at 20 weeks of age. **d** Serum proinsulin: insulin ratio (**e**) Heatmap showing mRNA fold change of insulin secretion and (**f**) insulin signaling markers, in whole pancreatic tissue, relative to control housekeeping gene *Ywhaz*. Box plots in (**b**–**d**) display the median (central line), 25th to 75th percentile (box) and minimum to maximum values (whiskers). Female animals; (**a**–**b**) $n = 12$ independent islets from 3 biologically independent animals (insulin), $n = 15$ independent islets from 3 biologically independent animals (WT, proinsulin), $n = 12$ independent islets from 3 biologically independent animals (*IL-22ra1*^ΔβKO, proinsulin). **c**, **d** $n = 8$ biologically independent wildtype (*IL-22ra*^fl/fl) and 9 biologically independent *IL-22ra1*^ΔβKO animals; **e**, **f** $n = 7$ biologically independent animals. Two-tailed unpaired Student's t-test; *$p < 0.05$, **$p < 0.01$; n.s., non-significant. *versus wildtype (*IL-22ra*^fl/fl) littermates. Scale bar: 20 um. Source data are provided as a Source Data file.

---

significantly elevated in the female *IL-22ra1*^ΔβKO animals compared to WT HFD animals (Supplementary Fig. 7a–c). Collectively, these results demonstrate that deficiency in beta-cell IL-22ra1 signaling impairs insulin biosynthesis. There is an increased secretion of proinsulin at the expense of mature insulin, when there is an increase in demand which results in impaired glycemic control.

### Endogenous pancreatic beta-cell IL-22ra1 signaling is required for appropriate islet growth and regeneration

Reduced insulin biosynthesis in the *IL-22ra1*^ΔβKO animals suggested impaired islet proliferation and potentially a reduction in beta-cell mass. Although male and female *IL-22ra1*^ΔβKO mice were generally lighter than their wild-type littermates, this difference was not significant (Supplementary Fig. 8a, c). Despite the absence of overt pancreatic exocrine or endocrine pathology, both male and female *IL-22ra1*^ΔβKO mice had lighter pancreata and smaller pancreatic islets than their wild-type counterparts (Fig. 4a, b; Supplementary Fig. 8b, d). Frequency distribution analyses of islet areas revealed a reduction in islet sizes and also suggested a reduction in islet numbers (Fig. 4c). Isolation of islets from *IL-22ra1*^ΔβKO animals then confirmed lesser absolute islet counts, compared to their wildtype littermates (Fig. 4d). To investigate the effect of IL-22ra1 signaling loss on transcriptional pathways, we examined the expression of islet regeneration and pancreatic growth genes. This analysis demonstrated that *Pax4*, *Neurod1*, *Neurog3* and *Reg3b*, genes key to pancreas development and islet regeneration were significantly downregulation in *IL-22ra1*^ΔβKO animals (Fig. 4e). The reduction in islet regeneration in the *IL-22ra1*^ΔβKO animals was mirrored by significantly fewer Ki-67+ cells per islet area, compared

to their wildtype littermates (Fig. 4f, g). Taken together, these results suggest that IL-22ra1 signaling is essential for the maintenance of islet growth and regeneration.

### Beta-cell dysfunction is accompanied by increased inflammation and cellular stress in the absence of IL-22ra1

Given that oxidative and ER stress can drive beta-cell dysfunction, leading to defects in insulin processing and secretion[11,23], we next examined if genes in the cellular stress pathways were altered. Significant increases in the key ER stress marker *sXbp-1*, and the antioxidant *superoxide dismutase 2 (Sod2)* were observed in the absence of beta-cell IL-22ra1 (Fig. 5a). Immunofluorescence staining confirmed the increase in the ER chaperone, Grp-78 (78-kDa glucose regulated protein, which is upregulated during ER stress[2,11]) in the *IL-22ra1*^ΔβKO animals (Fig. 5c). Moreover, a 2.5 fold increase in the oxidative stress marker, 4-hydroxynonenal (4-Hne), was noted in the islets of *IL-22ra1*^ΔβKO mice (Fig. 5d). A remaining question was whether there was a change in islet inflammation. Inflammation drives ER stress and the levels of systemic circulating inflammatory cytokines remained unaltered in the *IL-22ra1*^ΔβKO mice (Supplementary Fig.. 9). However, gene expression analysis revealed increased pancreatic inflammation, including the macrophage marker *Cd68* (Fig. 5b). Iba-1 staining (ionized calcium binding adaptor molecule 1; marker of macrophages) confirmed an increase in macrophage infiltration in *IL-22ra1*^ΔβKO animals (Fig. 5b, e). Interestingly, we also observed an increase in MHC II (major histocompatibility II) staining in islets of *IL-22ra1*^ΔβKO animals (Fig. 5f). Whilst this is not expressed on normal pancreatic beta-cells, previous studies have shown beta-cell MHC II expression in patients

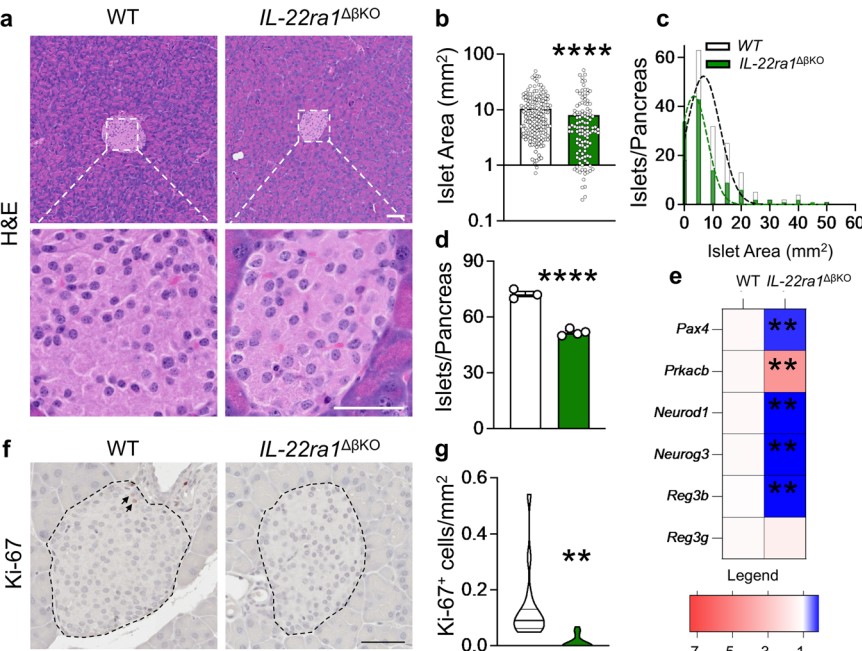

**Fig. 4 | Endogenous pancreatic β-cell IL-22ra1 signaling modulates islet growth and regeneration. a** H&E sections from pancreatic tissue. **b** Pancreatic islet area ($p < 0.0001$), and **c** frequency distribution of islet areas by size from serial pancreatic sections in animals at 20 weeks of age. **d** Absolute islet counts in animals following pancreatic islet isolation at 20 weeks of age ($p < 0.0001$). **e** Heatmap showing mRNA fold change of islet growth/regeneration markers in whole pancreatic tissue, relative to control housekeeping gene *Ywhaz*. **f** Immunohistochemical staining of pancreatic sections for Ki-67 at 8 weeks of age. **g** Number of Ki-67⁺ cells per islet area ($p = 0.0090$). All graphs are presented as Mean ± SEM. Female animals; **a–c** $n = 165$ independent islets (all islets in serial sections) from 3 biologically

independent wildtype (*IL-22ra1^fl/fl^*), and 115 independent islets (all islets in serial sections) from 3 biologically independent *IL-22ra1^ΔβKO^* animals. **d** $n = 3$ biologically independent wildtype (*IL-22ra1^fl/fl^*) and 4 biologically independent *IL-22ra1^ΔβKO^* animals. **f, g** $n = 16$ independent islets (all islets in one section) from 3 biologically independent wildtype (*IL-22ra1^fl/fl^*), and 10 independent islets (all islets in one section) from 3 biologically independent *IL-22ra1^ΔβKO^* animals. **e** $n = 7$ biologically independent animals; Two-tailed unpaired Student's t-test. ***p < 0.01, ****p < 0.0001; n.s., non-significant. * versus wildtype (IL-22ra1^fl/fl^) littermates. Scale Bar: 50 um. Source data are provided as a Source Data file.

with type I diabetes[24,25]. To confirm the increase in intra-islet MHC II staining observed in *IL-22ra1^ΔβKO^* mice was driven by the lack of IL-22ra1 signaling, we isolated and co-treated wildtype islets with mIFNγ and mIL-22-Fc. This revealed IL-22 treatment significantly reduced IFNγ-induced islet MHC II induction (Supplementary Fig. 10).

### Beta-cell IL-22ra1 signaling is required for appropriate glucose-stimulated insulin processing and secretion

We observed a reduction in total insulin but not proinsulin secretion in male animals at 20 weeks of age (Supplementary Fig. 11a–c), which collectively, could explain the less pronounced phenotype observed in male animals compared with female animals. In vivo insulin secretion is regulated by circulating/interstitial glucose, as well as the incretin glucagon-like peptide-1 (GLP-1), which is secreted postprandially. To further evaluate the role of IL-22ra1 in insulin secretion from beta-cells in isolation, we isolated islets from *IL-22ra1^ΔβKO^* animals. Islets were stimulated with physiological level of high (20 mM) and low (2.8 mM) glucose, and high glucose combined with GLP-1 (100 nM). Isolated islets from male and female *IL-22ra1^ΔβKO^* animals produced less total insulin than those from their littermates (Fig. 6a; Supplementary Fig. 11d). Additionally, in female *IL-22ra1^ΔβKO^* animals more of the total insulin was secreted as proinsulin, which was reflected by a significant increase in proinsulin: insulin ratios (Fig. 6b, c). The changes in proinsulin: insulin ratio suggests a decrease in the functional quality of insulin in the *IL-22ra1^ΔβKO^* animals. Therefore, we treated 3T3-L1 differentiated into adipocytes with equal amounts of total insulin (determined using ELISA) secreted by islets from WT littermate and *IL-22ra1^ΔβKO^* animals and subsequently measured the uptake of the fluorescent d-glucose analogue, 2-[N-(7-nitrobenz-2-oxa-1,3-diazol-4-yl) amino]-2-deoxy-D-glucose (2-NBDG). Mirroring the unaltered

proinsulin: insulin ratios in male *IL-22ra1^ΔβKO^* islets, 3T3-L1 adipocytes treated with total insulin from male *IL-22ra1^ΔβKO^* islets had similar 2-NBDG uptake compared to wildtype islet secretions (Supplementary Fig. 11f, g). However, insulin secreted from female *IL-22ra1^ΔβKO^* islets were found to have a significant reduction in 2-NBDG uptake compared to wildtype littermate islets (Fig. 6d), confirming the lower quality of insulin. In summary, we show here that selective ablation of pancreatic beta-cell IL-22ra1 signaling leads to age-related hyperglycaemia associated with compromised insulin biosynthesis, reduced islet proliferation, increased islet cellular stress, inflammation, and MHC-II expression (Fig. 7).

## Discussion

Whilst previous studies have explored the effects of exogenous IL-22 administration on insulin secretion, the role of endogenous IL-22ra1 signaling in pancreatic beta-cells is poorly understood. Here we report the occurrence of hyperglycemia and islet cell loss in mice lacking IL-22ra1 selectively in beta-cells of the pancreas. Loss of endogenous IL-22ra1 signaling in pancreatic beta-cells not only compromised total insulin biosynthesis but also impacted the quality of insulin secreted (increased proinsulin: insulin ratio) and the maintenance of islet/beta-cell mass with age.

The development of hyperglycemia with age in the *IL-22ra1^ΔβKO^* animals suggests that IL-22 signaling is protective and prevents the age-related decline of beta-cell function and number. Importantly, ageing is a risk factor for numerous metabolic disorders, including type 2 diabetes (T2D). Our data suggests that in mice there is a sex-dependent role for pancreatic beta-cell IL-22ra1 signaling in maintaining insulin biosynthesis and secretion, with this being more prominent in females compared to males. This is interesting as females

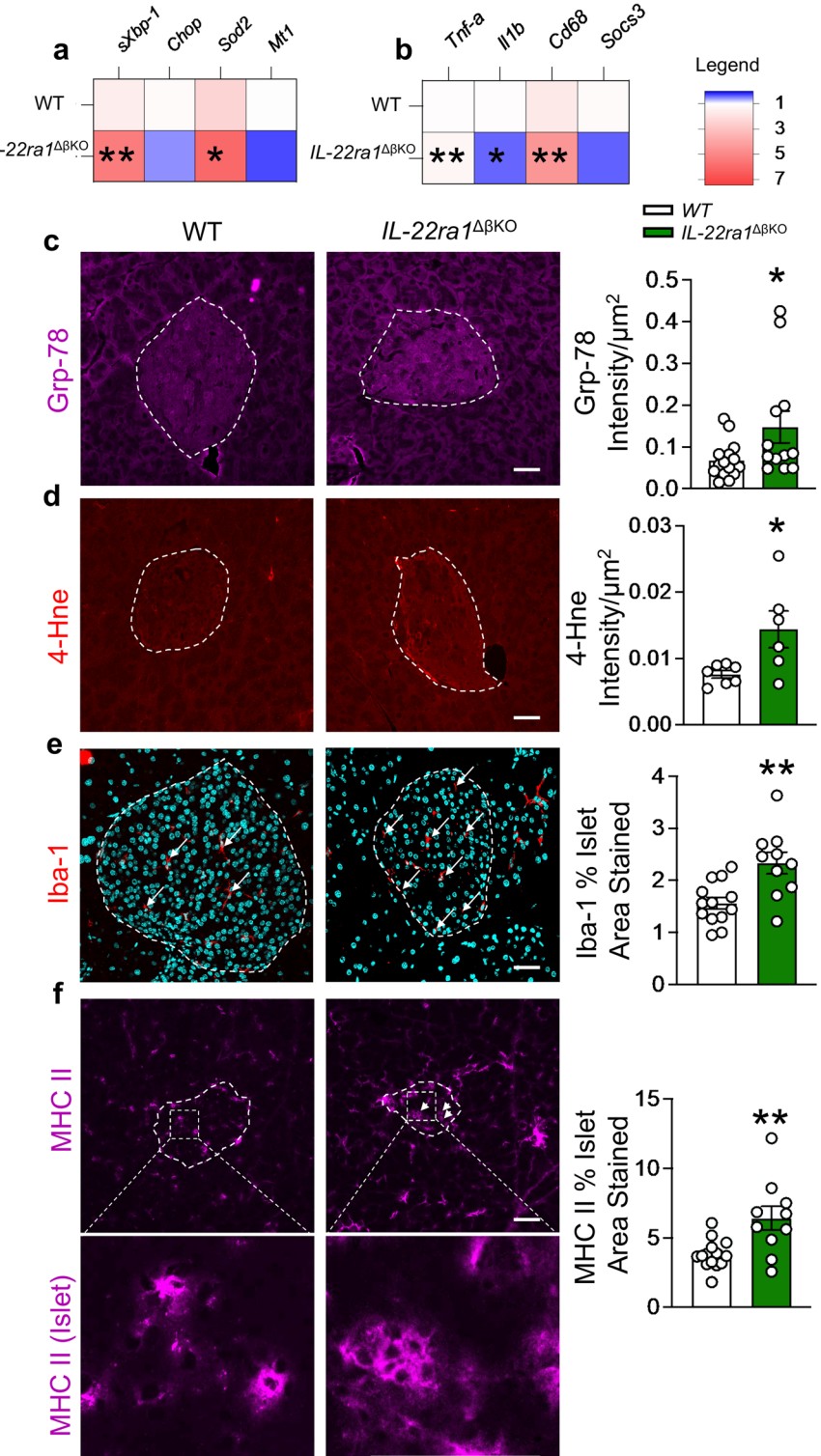

**Fig. 5 | Lack of endogenous pancreatic β-cell IL-22ra1 signaling causes islet dysfunction due to increased inflammation and cellular stress. a** Heatmap showing mRNA fold change in cellular stress markers and, **b** inflammatory markers in whole pancreatic tissue, relative to control housekeeping gene *Ywhaz*. **c** Mean intensity of Grp-78 per islet area (*p = 0.0386*) and representative image. **d** Mean intensity of 4-Hne per islet area (*p = 0.0252*) and representative image. **e** Percentage of islet area stained by Iba-1 (*p = 0.0024*) and representative image. **f** Percentage of islet area stained by MHC II (*p = 0.0034*) and representative image. All bar graphs are presented as Mean ± SEM. Female animals; **a**, **b** *n* = 7 biologically independent animals. **c** *n* = 15 independent islets (all islets in one section) from 3 biologically independent wildtype (*IL-22ra^fl/fl*), and 12 independent islets (all islets in one

section) from 3 biologically independent *IL-22ra1^ΔβKO* animals. **d** *n* = 7 independent islets (all islets in one section) from 3 biologically independent wildtype (*IL-22ra^fl/fl*), and 6 independent islets (all islets in one section) from 3 biologically independent *IL-22ra1^ΔβKO* animals. **e** *n* = 13 independent islets (all islets in one section) from 3 biologically independent wildtype (*IL-22ra^fl/fl*), and 10 independent islets (all islets in one section) from 3 biologically independent *IL-22ra1^ΔβKO* animals. **f** *n* = 14 independent islets (all islets in one section) from 3 biologically independent wildtype (*IL-22ra^fl/fl*), and 10 independent islets (all islets in one section) from 3 biologically independent *IL-22ra1^ΔβKO* animals. Two-tailed unpaired Student's t-test. **p* < 0.05, ***p* < 0.01; n.s., non-significant. *versus wildtype (IL-22ra1^fl/fl) littermates. Scale bar: 20 um. Source data are provided as a Source Data file.

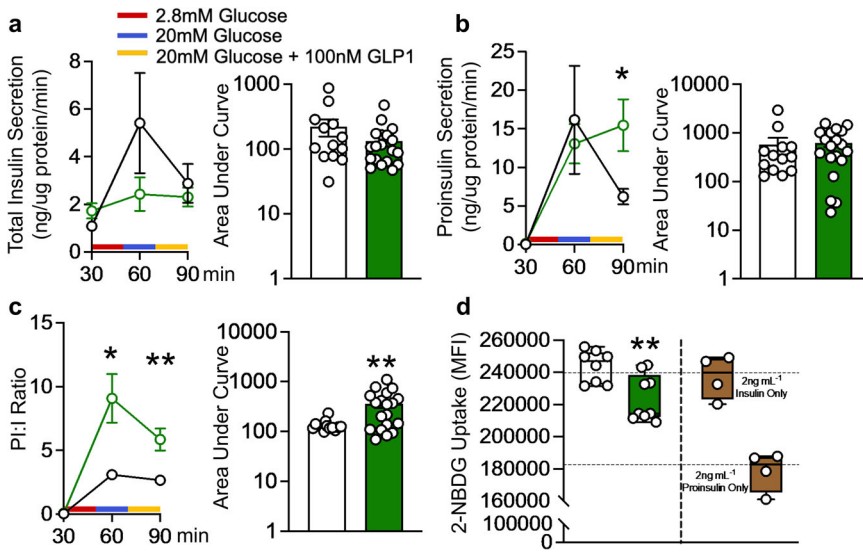

**Fig. 6 | Ablation of endogenous pancreatic β-cell IL-22ra1 signaling leads to poor quality insulin secretion and reduced glucose uptake. a** Total insulin secretion (ng/ug protein/min) (**b**) proinsulin secretion (ng/ug protein/min) (90 min; $p = 0.0459$), and **c** proinsulin: insulin ratio of mouse islets during in-vitro glucose stimulated insulin secretion, following stimulation with 2.8 mM glucose, 20 mM glucose and 20 mM glucose + 100 nM GLP-1. (60 min; $p = 0.0182$, 90 min; $p = 0.0056$, AUC; $p = 0.0076$). **d** 2-NBDG uptake in 3T3-L1 adipocytes exposed to 2 ng/mL islet insulin secretion following stimulation with 20 mM glucose + 100 nM GLP-1, $p = 0.0054$. All graphs in (**a-c**) are presented as Mean ± SEM, box plots in (**d**) display the median (central line), 25th to 75th percentile (box) and minimum to maximum values (whiskers). Female animals; **a**–**c** $n = 13$ independent samples (10 islets/sample) from 3 biologically independent wildtype (*IL-22ra^fl/fl*), and 19 independent samples (10 islets/sample) from 4 biologically independent *IL-22ra1^ΔβKO* animals. RM two-way ANOVA with the Geisser-Greenhouse correction and Sidak's multiple comparisons test (line graphs); two-tailed *Mann-Whitney* Test (bar/box plots). **d** $n = 8$ (*IL-22ra^fl/fl*) and 9 (*IL-22ra1^ΔβKO*) independent samples. *$p < 0.05$; n.s., non-significant; **$p < 0.01$; n.s., non-significant. *versus wildtype (*IL-22ra^fl/fl*) control. Source data are provided as a Source Data file.

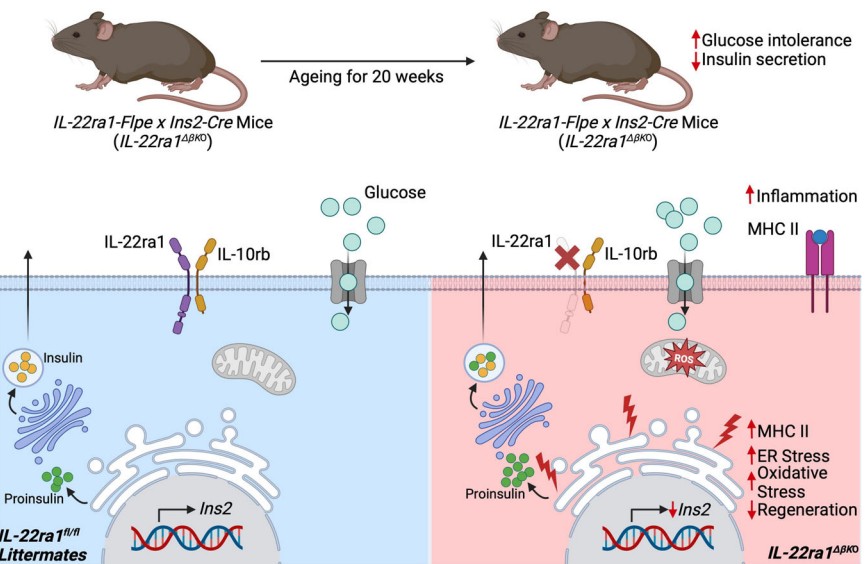

**Fig. 7 | Summary of findings.** Graphical abstract of findings showing that ablation of pancreatic beta-cell IL-22ra1 signaling leads to increased islet cellular stress and MHC II expression, reduced islet regeneration and insulin biosynthesis, and hypersecretion of proinsulin during glucose stimulation. These factors contribute to the age-related hyperglycaemia observed in *IL-22ra1^ΔβKO* animals. Figure created with BioRender.com, released under a Creative Commons Attribution-NonCommercial-NoDerivs 4.0 International license.

(both mice and humans) have previously reported to produce more IL-22[26]. Whilst there have been no reports of gender specific pancreatic islet *IL-22RA1* expression in humans, to understand these sex-specific differences, further studies are needed to investigate whether sex hormones like estrogen can regulate the endogenous levels of *IL-22ra1* or local IL-22 production in the islet.

Peripheral tissue insulin resistance, a hallmark of T2D, triggers compensatory hyperinsulinemia, placing a significant burden on beta-cells[18,27]. This increased workload manifests as cellular stress and is reflected by elevated proinsulin: insulin ratios in T2D patients[13,19]. Our study sheds light on a novel mechanism by which IL-22RA1 safeguards beta-cell function and glucose homeostasis during T2D progression. IL-22RA1 expression in human islets is increased in T2D and impaired glucose tolerance. Healthy human islets treated with anti-IL-22RA1 antibodies parallel alterations seen in islets of T2D patients (without changes to total insulin output). The targeted deletion of *IL-22ra1* in pancreatic beta-cells also precipitates age-dependent hyperglycemia, alongside heightened cellular stress and inflammation, without

inducing insulin resistance at 20 weeks of age. Intriguingly, in *IL-22ra1$^{\Delta\beta KO}$* animals, plasma glucose levels return to baseline post OGTT, suggesting that glucose normalization occurs independently of enhanced peripheral insulin sensitivity or response. Similarly in animal models, exogenous IL-22 administered ameliorated glycaemic control even before improvements in insulin sensitivity. This phenomenon suggests two potential and likely interconnected explanations. First, activation of insulin-independent glucose clearance mechanisms, possibly through upregulated activity of insulin-independent glucose transporters or alternative pathways that facilitate cellular glucose uptake, such as those stimulated by exercise or other physiological triggers. Second, this pattern could reflect the residual capacity of beta cells to eventually secrete enough insulin to manage glucose challenges, despite initial delays.

Pancreatic beta-cells have a large secretory load, producing a million proinsulin molecules per minute. To meet this demand, they require the unfolded protein response (UPR) signaling pathways, including XBP1, for insulin biosynthesis and secretion. In T2D patients and those at high risk for the disease, there is an increase in proinsulin: insulin ratio[28]. We have previously shown that IL-22 modulates the UPR in secretory cell types to promote protein biosynthesis[2,15,29]. Blocking IL-22 signaling in healthy human islets drives an increase in ER stress, oxidative stress, and proinsulin, similar to secretions from T2D islets. In *MIN6N8* beta-cells, RNA-seq analyses showed that IL-22 drove an antioxidant program. These data suggested an endogenous source of IL-22 modulates this pathway in the islets. Although IL-22 mRNA is increased in T2D islets, the source of local endogenous IL-22 remains to be determined. Our findings are also consistent with reports of *IL-22ra1$^{-/-}$* and female *IL-22$^{-/-}$* mice being more susceptible to diabetes[14,30]. However, some studies have also shown that *IL-22$^{-/-}$* knockouts do not show increased susceptibility to HFD-induced obesity[31]. Whilst IL-22RA1 forms a heterodimer with IL-10RB2 to facilitate IL-22 signaling, this receptor also binds to IL-20RB to form the IL-20 receptor type II (a receptor utilised by IL-20 and IL-24)[1]. This raises the possibility of an alternative IL-22RA1 ligand or redundancy between cytokines. Our cell-specific *IL-22ra1$^{\Delta\beta KO}$* and *IL-22ra1$^{\Delta\alpha KO}$* mice provide a better tool to understand IL-22ra1 biology, as IL-22ra1/IL-22RA1 expression is highest on pancreatic beta- and alpha-cells. Whilst no major changes in glucose control were observed in the *IL-22ra1$^{\Delta\alpha KO}$* mice, mirroring our in vitro findings, female *IL-22ra1$^{\Delta\beta KO}$* mice had increased pancreatic cellular stress. Additionally, stimulation of islets from these animals with glucose and GLP-1 resulted in a significant secretion of proinsulin. This supports the multiple reports of increased ER stress driving beta-cell dysfunction[13]. These data strongly suggest endogenous IL-22ra1 plays an integral part in regulating insulin quality through the modulation of intra-islet cellular stress.

Along with the low quality of insulin and increased cellular stress in the pancreatic beta-cells without IL-22ra1 signaling, we also observed reduced total insulin biosynthesis. This may stem from a suppression in islet growth and regeneration. There was a significant decrease in islet size and islet proliferation. At a molecular level, this was accompanied by reduced levels of *FoxO1*, *Neurod1*, *Mafa*, *Pdx1* and *Neurog3*. FoxO1 has previously been shown to preserve beta-cell mass in response to metabolic stress[32]. Activation of *FoxO1* is also known to lead to the increase in expression of the insulin-2 (*Ins2*) transcription factors *neurogenic differentiation 1 (Neurod1)* and *MAF BZIP Transcription Factor A (Mafa)*. Whilst FoxO1 is known to activate in response to oxidative stress to allow beta-cells to withstand acute metabolic stress, this mechanism cannot cope with chronic stress as indicated by a reduction in its expression in models of T2D[33]. Our findings support this idea; *IL-22ra1$^{\Delta\beta KO}$* animals had reduced pancreatic expression of *Neurod1* and *Mafa* following the development of progressively impaired glucose tolerance. These animals also had significant reductions in the master regulator of beta-cell fate *pancreatic and duodenal homeobox 1 (Pdx1)* and beta-cell differentiation gene *Neurogenin 3*

(*Neurog3*). These genes, together with *Mafa*, are known to be critical regulators of beta-cell development and regeneration[34]. The link with beta-cell regeneration is consistent with the well characterised role of IL-22 driving epithelial cell proliferation[35,36]. In particular, islet regeneration (Reg) proteins Reg 1 and 2, are C-type lectins expressed by beta-cells that activate cyclin D1 and promote beta-cell cycle progression and regeneration, which have been shown to be induced by IL-22. Moreover, neutralisation of IL-22 also led to suppression of Reg 1 and 2 regeneration genes in pre-diabetic models[37], supporting the role of IL-22ra1-signaling in inducing islet regeneration.

In *IL-22ra1$^{\Delta\beta KO}$* mice we observed an increase in pancreatic *Cd68* (macrophage-specific cell surface marker) and inflammatory cytokine mRNA expression, and this was verified by immunofluorescence for another marker (Iba1) demonstrating an increase in infiltrating macrophages in the islets. Pancreatic inflammation in *IL-22ra1$^{\Delta\beta KO}$* mice suggests a forward-feeding cycle potentially driven by increased islet oxidative and ER stress, leading to proinsulin misfolding and subsequent hyperglycemia. We have recently demonstrated that IL-22 suppresses mucosal epithelial cell MHC II[15] and we assessed whether the *IL-22ra1* signaling regulates beta-cell MHC II, which is proposed to increase in local islet inflammation in the context of Type 1 Diabetes[38]. Whilst inflammatory cytokines like IFNγ are known to upregulate beta-cell MHC II not much is known about pathways that regulate this process. Here, we show that IFNγ induced islet MHC II in healthy islets from wildtype animals, which was decreased by IL-22. Moreover, *IL-22ra1$^{\Delta\beta KO}$* mice have increased islet MHC II, suggesting this may be the reason there is an increase in local islet inflammation in these animals.

Impaired glycemic control is often associated with impaired gut mucosal barrier function and subsequent endotoxin associated systemic inflammation[39]. Whilst we did not examine gut barrier integrity in this study, interestingly, similar to *IL-22ra1$^{-/-}$* animals[15], no changes in systemic inflammation were noted in the *IL-22ra1$^{\Delta\beta KO}$* mice. However, it is important to note that most studies examining gut barrier integrity during impaired glucose metabolism involve obesity or other forms of systemic metabolic stress[29,40,41]. We show that *IL-22ra1$^{\Delta\beta KO}$* mice had impaired glucose tolerance even in the absence of insulin resistance and obesity.

Pancreatic alpha cells play a key role in maintaining glucose homeostasis and are also known to express IL-22RA1[42]. However, in this study we show that reveal that alpha-cell-specific IL-22ra1 knockout (*IL-22ra1$^{\Delta\alpha KO}$*) does not impact key metabolic parameters in our mouse model. Specifically, these animals demonstrated no significant differences in body or pancreas weights, serum glucagon levels, glucose tolerance, insulin sensitivity, or endocrine/exocrine pathology when compared to littermate controls. Nonetheless, it is important to consider the potential roles of IL-22ra1 signaling in alpha cells that may not have been captured in the scope of our study. IL-22 is known for its role in tissue protection and regeneration, and whilst our data does not demonstrate a direct effect on the parameters measured, IL-22ra1 could potentially influence alpha cell stress responses, contribute to the maintenance of alpha cell mass, or play a role in intra-islet signaling that impacts the local islet microenvironment. Further research is necessary to elucidate these potential roles of IL-22RA1 in alpha cells, particularly under different physiological or pathological conditions that were beyond the scope of the current investigation.

Overall, our findings highlight the importance of IL-22ra1 signaling, suggesting that IL-22ra1 pancreatic beta-cell signaling is important in the context of T2D, and it also protects beta-cells from an age-related decline in function. IL-22ra1 signaling maintains beta-cell regeneration, proliferation and insulin biosynthesis and secretion. Complementarily, recent work has also demonstrated that targeted IL-22 therapy can significantly enhance pancreatic function and insulin response, further validating the therapeutic potential of this pathway in metabolic diseases[43]. Given these findings, greater understanding of the role of endogenous IL-22RA1 signaling is of clinical

importance, with potential therapeutic avenues in manipulating this signaling pathway to preserve beta-cell health and function.

## Methods

### Human pancreatic islet isolation and culture

For human islet experiments in Fig. 1, we obtained approval for procuring and performing experiments with human islets from the Mater Health Services Human Research Ethics Committee (ER Stress in Pancreatic Islet – HREC/MML/23899). We obtained human pancreatic islets from 3 healthy and 3 T2D organ donors (Demographics in Table S3) via the Tom Mandel Islet Transplant Program in Australia and Prodo laboratories. Consent for use for research was given by the relatives of the donors.

At the Tom Mandel Islet Transplant Program, we prepared islets within a fully contained Isolator (BioSpherix Xvivo System, Lacona, NY) situated in a clean room facility and using a variation of an established method. We removed pancreata from heart-beating deceased donors and disaggregated them by infusing the ducts with cold collagenase (SERVA, Heidelberg, Germany). Dissociated islet and acinar tissue were then separated on a continuous Biocoll (Biochrom AG, Berlin) density gradient on a refrigerated apheresis system (Model 2991, COBE Laboratories, Lakewood, CO). Next, we counted purified islets and expressed islet number and mass in terms of islet equivalents (IEQ)66. Islets were transported as soon as possible after isolation and received 1–7 days post-isolation. On arrival, islets were immediately placed into culture in CMRL (5.5 mM glucose) containing 10% FCS and 50 U mL$^{-1}$ penicillin and streptomycin (Life Technologies). Purity of the islet preparations were between 70–99%. Following overnight culture, individual islets free from exocrine tissue were handpicked and established in culture with an equivalent number and size distribution for experiments.

Following overnight culture, human islets were plated at 10 islets per well in 5.5 mM glucose and treated them with 10 µg mL$^{-1}$ anti-IL-22RA1 (Clone 496514; R&D). We conducted static glucose-stimulated insulin secretion tests by challenging islets consecutively with 2.8 mM glucose for 30 min, 20 mM glucose for 30 min and 100 nM GLP-1 (7–36 active peptide; Sigma-Aldrich) in 20 mM glucose for a further 30 min. We collected supernatants for insulin and proinsulin secretion measurements, and the islets were homogenized in Trizol for RNA analysis.

### Laser capture microdissection (LCM) of human pancreatic islets

The study protocol for human islet experiments in Supplementary Fig. 1 (ClinicalTrials.gov registration no. NCT02175459), was approved by the local ethics committee (P/656/CE2010 and 22573/14) (Rome, Italy) and all participants provided written informed consent, which was followed by a comprehensive medical evaluation.Twelve patients (Demographics in Table S3) undergoing pylorus-preserving pancreatoduodenectomy were recruited from January 2017 to July 2019 at the Digestive Surgery Unit and studied at the Centre for Endocrine and Metabolic Diseases unit (Agostino Gemelli University Hospital, Rome, Italy). All patients underwent complete metabolic phenotyping including OGTT and mixed meal test (MMT). As previously described[13], participants were metabolically profiled prior to surgery. Based on thresholds set by the ADA for fasting glucose, HbA1c and 2 h glucose level during an OGTT in the days immediately before surgery, participants were then classified as NGT ($n = 4$), IGT ($n = 4$) or with disease onset longer than 1 year; T2D ($n = 4$).

Pancreatic human tissue samples from NGT, IGT and T2D donors were frozen in Tissue-Tek OCT compound (Sakura Finetek Europe, the Netherlands), and 7-µm-thick sections were cut from frozen OCT blocks. Sections were fixed in 70% Ethanol for 30 s, dehydrated in 100% Ethanol for 1 min, in 100% Ethanol for 1 min, in Xylene for 5 min and finally air-dried for 5 min. Laser capture microdissection (LCM) was performed using the Arcturus XT Laser-Capture Microdissection system (Arcturus Engineering, Mountain View, CA, USA)Human pancreatic islets were subsequently visualised through beta cell autofluorescence and captured using CapSure™ HS LCM Caps (Thermo-Fisher Scientific, Waltham, MA, USA) and infrared (IR) laser. Adhesive thermoplastic caps containing microdissected cells were incubated with 10 µl of Extraction Buffer (kit0204-ThermoFisher Scientific, Waltham, MA, USA) for 30 min at 42 °C and kept at −80 °C until RNA extraction. Each microdissection was performed within 30 min from the staining procedure in a contamination-free-dehumidified environment with an external temperature of 16 °C to preserve RNA integrity. Overall $n = 50$ microdissected pancreatic islets per case were analysed for the molecular analysis.

### RNA extraction from LCM isolated islets and quality control

Total RNA was extracted from each LCM sample using PicoPure RNA isolation kit Arcturus (kit0204-ThermoFisher Scientific, Waltham, MA, USA) following manufacturer's procedure. Briefly, the cellular extracts were firstly mixed with 12.5 µl of 100% Ethanol and then transferred onto the purification column filter membrane. DNase treatment was performed using RNase-Free DNase Set (Qiagen, Hilden, Germany). Total RNA was finally eluted in 11 µl of Elution Buffer (DNase/RNase-Free Water). All LCM captures deriving from the same human sample were pooled and subjected to a subsequent concentration through Savant SpeedVac™ SC100 centrifugal evaporator.

In order to evaluate RNA abundance and purity, Agilent 2100 Bioanalyzer technology with RNA Pico chips (cat. 5067-1513 Agilent Technologies, Santa Clara, CA, USA) was performed for each RNA sample reporting RNA integrity (RIN) and concentration and by excluding samples with RIN < 5.0.

### Gene expression analysis through droplet digital PCR (ddPCR)

In order to analyse gene expression, a reverse transcriptase reaction was performed on each RNA sample extracted (500 pg) from microdissected islet from each donor using SuperScript™ VILO™ cDNA Synthesis Kit (cat. 11754050-ThermoFisher Scientific, Waltham, MA, USA). cDNA deriving was then amplified using TaqMan PreAmp Master Mix (cat. 4488593, ThermoFisher Scientific, Waltham, MA, USA) following manufacturer's instructions. Then, droplet digital PCR (ddPCR) was performed on a BioRad QX200 system using a Probes assay (BioRad, Mississauga, ON, Canada). Each PCR reaction contained 11 µL of QX200 2x ddPCR Supermix for probes (no dUTP), 1.1 µL of each 20X TaqMan assay, 5.9 µL of H2O and 4 µL of template cDNA in a final volume of 22 µL. The PCR reactions were mixed, centrifuged briefly and 20 µL transferred into the sample well of a DG8™ cartridge. After adding 70 µL of QX200™ droplet generation oil into the oil wells, the cartridge was covered using a DG8™ gasket, and droplets generated using the QX200™ droplet generator. Droplets were carefully transferred into PCR plates using a multi-channel pipette and the plate sealed using PCR plate heat seal foil and the PX1™ PCR plate sealer. PCR was performed in a C1000 touch thermal cycler (BioRad, Mississauga, ON, Canada). The PCR protocol was 95 °C for 10 min; 40 cycles of: 95 °C for 30 s and 56 °C for 1 min; 98 °C for 10 min and 4 °C for 30 min. PCR plates were transferred into a QX200™ droplet reader to count positive and negative droplets. Thresholds to separate positive from negative droplets were set manually for each gene using the histogram function and reads analysed using QuantaSoft™ Analysis Pro software (Version 1.2, BioRad, Mississauga, ON, Canada).

### Animal experiments

All mice were housed in sterilized, filter-topped cages in a conventional clean facility. Animals were maintained on a 12:12-h light-dark cycle, at ambient temperature (22 - 23 °C), 40–60% humidity and received food (specific diets specified below) and water *ad libitum* with nesting materials. All experiments were approved by the University of Queensland Animal Ethics Committee (AE519/16, 2021/AE000426) and conducted in accordance with guidelines set out by the National

Health and Medical Research Council of Australia. *IL-22ra1-FLPe* (B6.Cg-*Il22ra1^{tm1.1Koll}*/J; Jackson Laboratory Strain #:031003) animals were crossed with either *Ins2-cre* (B6.Cg-Tg(Ins2-cre)25Mgn/J; Jackson Laboratory Strain #:003573) or *Gcg-cre* (B6.Cg-Tg(Gcg-cre)1Herr/Mmnc/J; MMRCC, 000358-UNC) animals to generate *Ins2 cre x IL-22ra1-FLPe* and *Gcg cre x IL-22ra1-FLPe* colonies respectively.

8-weeks old male and female *Ins2 cre x IL-22ra1-FLPe* mice were fed *ad libitum* a lard high fat diet (HFD-lard; Speciality feeds Australia, SF04-001) or normal chow diet (NCD; Speciality feeds Australia, SF00-100) containing less than 10% saturated fat. 8-weeks old male and female *Gcg cre x IL-22ra1-FLPe* mice were fed *ad libitum* a normal chow diet (NCD; Speciality feeds Australia, SF00-100) containing less than 10% saturated fat (Speciality feeds). Metabolic tolerance tests were performed at 4-week intervals as indicated in the figures. On conclusion of each experiment, animals were euthanised via cervical dislocation, with the absence of hind foot pinch response confirming successful euthanasia.

## Murine pancreatic islet isolation and culture
*IL-22ra1^{ΔβKO}* and their wildtype littermates (*IL-22ra1^{fl/fl}*) were euthanised and freshly prepared collagenase P (Roche, Indianapolis, IN) solution (0.5 mg mL$^{-1}$) was injected into the pancreas via the common bile duct. The perfused pancreas was then digested at 37 °C for 20 min, and islets were purified via centrifugation on a Histopaque-1077 (Sigma; #10771) gradient. Islets were then handpicked under a stereoscopic microscope, enumerated, and cultured overnight in 5 mM glucose.

Following overnight culture, human islets were plated at 10 islets per well in 5.5 mM glucose. We conducted static glucose-stimulated insulin secretion tests by challenging islets (mouse and human) consecutively with 2.8 mM glucose for 30 min, 20 mM glucose for 30 min and 100 nM GLP-1 (7–36 active peptide; Sigma-Aldrich) in 20 mM glucose for a further 30 min. We collected supernatants for insulin and proinsulin secretion measurements, and the islets were homogenized in Trizol for RNA analysis. A subset of wildtype islets was also stimulated with 10 ng mL$^{-1}$ mIFNγ (R&D; Cat # 485-MI) in the presence/absence of 50 ng mL$^{-1}$ mIL-22-Fc for 48 h.

## Metabolic measurements
For oral glucose tolerance test (oGTT), mice were fasted for 5 h and then challenged with 50 mg of D-Glucose solution by oral gavage. Blood glucose levels were measured using a glucometer (SensoCard, 77 Electronika, Hungary) via tail bleeding at 0, 15, 30, 60, 120 min timepoints post-challenge. For i.p. insulin tolerance tests (ipITT), mice were fasted for 5 h, and then challenged with 0.75 U per kg body weight of insulin (Humalog, Lilly). Blood glucose levels were then measured using glucometer via tail bleeding at 0, 15, 30, 60, 120 min time-points post-challenge.

## qRT-PCR
qRT-PCR was performed according to the protocol described previously[2]. Briefly, cells were first lysed using TRIzol (Invitrogen). Pure RNA was then isolated using the ISOLATE II RNA Mini Kit from Bioline (Alexandria). For tissue samples, snap-frozen tissues were first homogenized using beads (Lysing Matrix D Bulk; MP Biomedicals) in TRIzol. RNA was then isolated as per manufacturer's instructions using the ISOLATE II RNA Mini Kit from Bioline (Alexandria). 1 µg of RNA was then used to synthesize cDNA, using the Bioline cDNA synthesis kit containing oligo (dT) and random hexamers. All cDNA was then diluted in a 1:10 ratio to perform qRT-PCR.

2.5 µL of diluted cDNA, 0.75 µL of desired primer (2 µM), 3.75 µL of SYBR green (SensiFAST SYBR Lo-ROX kit; Bioline), and 0.5 µL of DNase and RNase free water were mixed and measured on a Real-Time PCR System (Applied Biosystems ViiA 7; Life Technologies Corporation) for 40 cycles. The resulting Ct values were then analysed using the ViiA 7 software (Life Technologies Corporation). The relative expression of target genes was determined by the ΔΔCt method and normalized to housekeeping gene *GAPDH* or *Ywhaz* and expressed as a fold difference to the mean of the relevant control samples. See tables S1 (human) and S2 (mouse) for details on primers used in this study.

## RNA-Seq
We performed next-generation sequencing in *MIN6N8* mouse insulinoma cells treated with PBS or IL-22 (50 ng mL$^{-1}$, 4 h). mRNA sequencing was performed by the Australian Genome Research Facility using HiSeq 2500 machine (Illumina) with a maximal read length of 100 bp. Differential gene expression analysis was performed using the Qiagen Ingenuity Pathway Analysis (IPA) Software.

## ELISA
Human Insulin (Mercodia; Cat # 10-1113-01), human proinsulin (Mercodia; Cat # 10-1118-01), mouse insulin (Crystal Chem; Cat # 90080), mouse proinsulin (Mercodia; Cat # 10-1232-01), mouse glucagon (Crystal Chem; Cat # 81518), mouse LEGENDplex custom 10-plex (Biolegend) were measured using ELISA kits following manufacturer's instructions.

## Immunofluorescence
6 µm thick slides were cut from paraffin embedded pancreas blocks, deparafinised, rehydrated and heat treated in 10 mM citrate buffer (pH 6.0) at 95 °C for 35 min prior to staining. Following incubation in 10% KPL (Seracare) for 30 min, slides were incubated the following primary antibodies overnight at 4 °C: Rabbit anti-Iba1 (Novachem, Cat # S03866; 1:1000), Rat anti-IL-22ra1 (R&D, Cat # MAB-42941; 1:100), Mouse anti-proinsulin (R&D, Cat # MAB-13361; 1:200), Guinea Pig anti-insulin (Invitrogen, Cat # PA-26938; 1:500), Rabbit anti-Grp78 (Sigma Aldrich, Cat # 8918; 1:1000), Mouse anti-4-Hne (Invitrogen, Cat # MA5-27570, 1:200), Rat anti-MHC II (Invitrogen, Cat # 14-5321-82; 1:500), Rabbit anti-Ki-67 (Invitrogen, Cat # MA5-14520; 1:500), Rabbit anti-Cre recombinase (Cell Signaling Technology, Cat # 15036; 1:200). They were then stained with the respective fluorescent secondary antibodies against AF488 Goat anti-Guinea Pig (Invitrogen, Cat # A-11073; 1:500), AF555 Goat anti-Rabbit (Invitrogen, Cat # A-32732; 1:500), AF647 Goat anti Rabbit (Invitrogen, Cat # A-32733, 1:500), AF555 Goat anti Rat (Invitrogen, Cat # A-21434; 1:500), AF555 Goat anti Mouse (Invitrogen, Cat # A-21422; 1:500) for 60 min, counter stained with heochyst 33342 stain (Invitrogen; Cat # 62249), and mounted. Slides then were imaged on the Olympus FV3000 confocal microscope. Image quantification was performed via ImageJ, where target antigens area stained/intensity was measured relative to islet area.

## Cell culture
We maintained *MIN6N8* cells (a kind gift from J. Miyazaki, Osaka University) in phenol-red–free DMEM (Life Technologies) containing 25 mM glucose (3.4 g L$^{-1}$ sodium bicarbonate, 50 U mL$^{-1}$ penicillin and streptomycin, 71.5 µM β-mercaptoethanol and 10% heat-inactivated FBS) and transferred them to DMEM as above with 5.5 mM glucose 48 h before experimentation. 3T3-L1 cells were obtained from the American Type Culture Collection (ATCC, Manassas, Virginia, USA; CL-173), and maintained in DMEM (Life Technologies) media containing 25 mM glucose (3.4 g L$^{-1}$ sodium bicarbonate, 50 U mL$^{-1}$ penicillin and streptomycin and 10% heat-inactivated FBS). All cells were maintained in a humidified atmosphere of 5% CO2 in the air at 37 °C.

## Glucose uptake assay
We differentiated 3T3-L1 cells into adipocytes, by stimulating $1 \times 10^4$ cells in a black 96-well plate with DMEM (Life Technologies) media containing 25 mM glucose (3.4 g L$^{-1}$ sodium bicarbonate, 50 U mL$^{-1}$ penicillin and streptomycin and 10% heat-inactivated FBS), 0.5 mM methylisobutylxanthine (IBMX; Sigma), 1 µM dexamethasone (Sigma) and 10 µg mL$^{-1}$ insulin (Sigma) for 2 days. This media was then replaced

with fresh media containing 10 µg mL$^{-1}$ insulin for a further 2 days. Following this, media was replaced with fresh media every 48 h, until adipocyte differentiation.

All experiments were carried out on day 8 post-differentiation. Cells were cultured in serum-free 2.8 mM glucose media for 2 h before the addition of insulin. We treated cells for 1 h with 2 ng ml$^{-1}$ recombinant insulin, proinsulin, or total insulin from islet secretions at 37 °C as described in figure legends. Subsequently, cells were incubated with 10 µM 2-(N-(7-nitrobenz-2-oxa-1,3-diazol-4-yl) amino)-6-deoxyglucose (2-NBDG; Molecular Probes) for 20 min before washing extensively with PBS and determining glucose uptake using a POLARstar Omega plate reader.

## Statistical analysis

The in vivo experiments were powered for a 1.5 s.d. change in the area under the curve of the glucose tolerance test as the major primary outcome measure. Statistical analyses were performed using GraphPad PRISM version 9.01 (GraphPad software, Inc.) as described in individual figure legends. After confirmation of a normal distribution by probability plots, differences between groups were assessed by using parametric tests (one-way ANOVA with post-test or, where appropriate, a two-tailed Student t-test). Where a normal distribution could not be confirmed, non-parametric tests were used (Kruskal-Wallis non-parametric ANOVA with a Dunn's post hoc test or, where appropriate, a Mann-Whitney U test).

## Reporting summary

Further information on research design is available in the Nature Portfolio Reporting Summary linked to this article.

## Data availability

The RNA-Seq dataset generated and analysed during the current study is available on the NCBI Gene Expression Omnibus (GEO) database, under accession code GSE262867. All other data generated in this study are provided in the Supplementary Information/Source Data file. Source data are provided with this paper.

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

## Acknowledgements

We would like to thank the staff of the Mater Research and Translational Research Institute Biological Research Facilities for care of breeding and experimental animals. The authors thank Thomas Loudovaris, Thomas W Kay and Helen E Thomas (St. Vincent's Research Institute, Melbourne, Victoria, Australia) for providing the human pancreatic islet samples. H.S was supported by the Australian Government Research Training Program (RTP) Scholarship, the Mater Research Frank Clair Scholarship, a Gastroenterological Society of Australia Project Grant (S.Z.H; H.S), and Australian and a New Zealand Society for Immunology Postgraduate Career Advancement Grant (H.S). S.K was supported by Diabetes Australia Research Grant. This work was supported by Australian National Health and Medical Research Council (NHMRC) Ideas Grant (APP1183713, G.A.R; G.A.M; J.B.P; M.A.M; S.Z.H), Career Development Fellowship (S.Z.H), Gastroenterological Society of Australia (S.Z.H; G.A.M) and Mater Foundation (S.Z.H). This study was also supported by grants from the Università Cattolica del Sacro Cuore (Fondi Ateneo Linea D.1, anno 2019, and Fondi Ateneo Linea D.1, anno 2020); the Italian Ministry of Education, University, and Research (MIUR) (GR-2018-12365577 to T.M, RF-2019–12369293 to A.G and PRIN 2020SH2ZZA to A.G).

## Author contributions

Conceptualization, S.Z.H; Methodology, H.S, K.Y.W, and S.Z.H; Investigation, H.S, K.Y.W, A.M, S.K, E.A, T.M, A.G, G.S, F.D, and S.Z.H; Formal Analysis, H.S, and S.Z.H; Writing – Original Draft, H.S, and S.Z.H; Writing – Reviewing & Editing, H.S, K.Y.W, A.M, S.K, L.B, E.A, T.M, A.G, G.S, F.D, G.A.R, G.A.M, M.A.M, J.B.P, and S.Z.H; Funding Acquisition, L.B, G.A.R, G.A.M, J.B.P, M.A.M, S.Z.H, T.M, and A.G; Supervision, S.Z.H.

## Competing interests

S.Z.H, M.A.M and J.B.P are inventors on a patent relating to IL-22 use in metabolic disease. The remaining authors declare no other competing interests.

## Additional information

**Peer review information** : *Nature Communications* thanks Ernest Adeghate and the other, anonymous, reviewer(s) for their contribution to the peer review of this work. A peer review file is available.

¹Immunopathology Group, Mater Research Institute-The University of Queensland, Translational Research Institute, Brisbane, QLD, Australia. ²Faculty of Medicine, The University of Queensland, Brisbane, QLD, Australia. ³Department of Respiratory and Sleep Medicine, Mater Health, South Brisbane, QLD, Australia. ⁴Diabetes Unit, Department of Medicine, Surgery and Neurosciences, University of Siena, Siena, Italy. ⁵Dipartimento di Medicina e Chirurgia Traslazionale, Università Cattolica del Sacro Cuore, Roma, Italy. ⁶Pancreas Unit, CEMAD Centro Malattie dell'Apparato Digerente, Medicina Interna e Gastroenterologia, Fondazione Policlinico Universitario Gemelli IRCCS, Roma, Italy. ⁷Endocrinology and Diabetology Unit, Fondazione Policlinico Universitario Gemelli IRCCS, Roma, Italy. ⁸Tuscany Centre for Precision Medicine (CReMeP), Siena, Italy. ⁹QIMR Berghofer Medical Research Institute, Brisbane, QLD, Australia. ¹⁰Department of Gastroenterology and Hepatology, Princess Alexandra Hospital, Brisbane, QLD, Australia. ¹¹School of Medicine, Dentistry and Health Sciences, University of Melbourne, Parkville, VIC, Australia. ¹²Health Translation Queensland, Royal Brisbane and Women's Hospital, Herston, QLD, Australia. ¹³Australian Infectious Disease Research Centre, University of Queensland, Brisbane, QLD, Australia. ✉e-mail: sumaira.hasnain@mater.uq.edu.au

