## [Peer Review File · Nature Communications]

Pancreatic Beta-Cell IL-22 Receptor Deficiency Induces Age-Dependent Dysregulation of Insulin Biosynthesis and Systemic Glucose HomeostasisREVIEWER COMMENTS

Reviewer #1 (Remarks to the Author):

The manuscript entitled “Pancreatic Beta-Cell IL-22 Receptor Deficiency Induces Age-Dependent Dysregulation of Insulin Biosynthesis and Systemic Glucose Homeostasis” provides convincing evidence that IL22 prevents insulin deficiency by regulating multiple activities in pancreatic beta cells. The authors have shown that IL22 signaling maintains UPR, promotes islet growth and regeneration, and reduces inflammatory response. This work provides mechanistic insights into the important regulatory role of IL22 in beta cells, which should be interesting to a broad range of Nature Communication’s readers. However, the following issues need to be addressed or discussed.

(1) MIN6N8 cells were used for RNA-seq analysis. MIN6N8 cells are a mixed cell line that includes pancreatic beta cells and other pancreatic cells. Therefore, the result may not accurately represent the transcriptional profile in pancreatic beta cells. In addition, dissected islets were used for ddPCR analysis, which could not show the transcriptional changes in pancreatic beta cells. To address these issues, it is better to perform single cell RNA-seq on isolated islet cells.

(2) Interestingly, the authors showed that defective IL22 signaling increasing ER stress, oxidative stress and proinsulin/insulin ratio similar to the changes found in the secretions from T2D islets. However, the authors have not discussed whether the changes in the secretions from T2D islets are the results of insulin resistance in T2D. This needs to be clarified to help understand possible relevance between the regulatory role of IL22 in beta cells and T2D, which has not been clearly addressed in this manuscript.

(3) In lines 220 and 221, the authors claim, “Perhaps the endogenous protection by IL-22RA1 signaling could contribute to the lower rates of beta-cell dysfunction in females compared to males”. This statement appears to be misleading. The data shown in this manuscript indicates that the mice with beta cell specific IL-22ra1 deficiency develop glucose intolerance with age, which was more pronounced in female mice. This means that compared to male mice, female mice’s glucose tolerance is more dependent on IL22

signaling in beta cells. Therefore, this evidence cannot help explain why women have a lower incidence of T2D compared to men. A weakness of this manuscript is that the clinical relevance is unclear.

(4) The authors have shown that IL22ra1 deficiency in pancreatic beta cells results in insufficient UPR, reduced islet growth and regeneration, and increased inflammatory cell infiltration. However, it is not clear which one is the main cause of glucose intolerance in beta cell specific IL22ra1 deficient mice.

(5) In lines 275-276, the authors explain that the ablation of IL-22ra1 may enhance beta-cell MHC II, leading to an increase in local islet inflammation. To support this view, the authors need to show increased MHC II in IL-22ra1 deficient beta cells.

Reviewer #2 (Remarks to the Author):

The manuscript entitled “Pancreatic Beta-Cell IL-22 Receptor Deficiency Induces Age-dependent Dysregulation of Insulin Biosynthesis and Systemic Glucose Homeostasis” by Sajiir et al., examined the effect of IL-22 Receptor ablation on glucose metabolism. The authors used a large variety of molecular, cellular, and morphological techniques to show that mice lacking the IL-22 receptor A1 have reduced insulin secretion, quality, and poor islet regeneration in addition to increased islet cellular stress and inflammation.

Comments

Line 30: What does “high-quality” insulin mean?

Line 50: “ER’ should be defined at first use

Line 101: The authors indicate that “... IL-22 treatment downregulated disease pathways...”. What are these pathways?

Lines 111-112: The author observed that “...animals developed increasingly severe glucose intolerance with age, which was more pronounced in female animals”. The authors should explain this difference in more detail.

Line 169: “Antioxidant” is written in 2 formats. It would be nice to use the same format

Line 210: Please delete “(Fig 7)” from the discussion

Line 364: Vendor details of the HFD should also be provided

Line 410: Please replace “Histology” with “Immunofluorescence”

Line 418: Which antibodies? At what dilution?

Lines 478 & 509: Refs 2 & 17 are the same

Other Comments

1. The authors should explain the possible role of IL-22RA on the plasma membrane of pancreatic alpha cells
2. Why does the glucose level tail down to normal at 120 min after the glucose challenge in Figure 2c?

We would like to thank all the reviewers for their insightful comments, which helped strengthen the manuscript significantly. We have addressed all the points raised as indicated below and altered the manuscript to incorporate all the new data/information where possible below as author responses (AR).

Reviewer #1 (Remarks to the Author):

The manuscript entitled “Pancreatic Beta-Cell IL-22 Receptor Deficiency Induces Age-Dependent Dysregulation of Insulin Biosynthesis and Systemic Glucose Homeostasis” provides convincing evidence that IL22 prevents insulin deficiency by regulating multiple activities in pancreatic beta cells. The authors have shown that IL22 signaling maintains UPR, promotes islet growth and regeneration, and reduces inflammatory response. This work provides mechanistic insights into the important regulatory role of IL22 in beta cells, which should be interesting to a broad range of Nature Communication’s readers. However, the following issues need to be addressed or discussed.

1. MIN6N8 cells were used for RNA-seq analysis. MIN6N8 cells are a mixed cell line that includes pancreatic beta cells and other pancreatic cells. Therefore, the result may not accurately represent the transcriptional profile in pancreatic beta cells. In addition, dissected islets were used for ddPCR analysis, which could not show the transcriptional changes in pancreatic beta cells. To address these issues, it is better to perform single cell RNA-seq on isolated islet cells.

AR01: We acknowledge that studying MIN6N8 cells and islets with multiple cell types has limitations for understanding beta-cell specific changes. Whilst single-cell RNA sequencing or high-resolution spatial transcriptomics would be ideal this is beyond our financial scope due to budgetary constraints. However, to ensure the robustness of our findings, we complemented the MIN6N8/human islet data with results from follow-up experiments using alpha or beta cell-specific IL-22ra1 knockouts. This two-pronged approach balances financial feasibility with scientific rigor, allowing us to gain valuable insights into beta-cell biology within budgetary constraints.

2. Interestingly, the authors showed that defective IL22 signaling increasing ER stress, oxidative stress and proinsulin/insulin ratio similar to the changes found in the secretions from T2D islets. However, the authors have not discussed whether the changes in the secretions from T2D islets are the results of insulin resistance in T2D. This needs to be clarified to help understand possible relevance between the regulatory role of IL22 in beta cells and T2D, which has not been clearly addressed in this manuscript.

AR02: Thank you to the reviewer for highlighting this. The text is now altered in the results section (line 84-91) and discussion (line 250) to highlight that in T2D, peripheral tissue insulin resistance can lead to compensatory increased insulin demand, where pancreatic beta cells increase insulin production to overcome resistance. During this compensatory process, as previously demonstrated in a model of beta cell mass reduction and increased beta-cell workload, circulating levels of proinsulin: insulin ratio significantly increase in subjects with insulin resistance compared to control (PMID: 30131390; PMID: 33905373). Further, we previously demonstrated that increased proinsulin/insulin ratio in situ is often increased together with ER and oxidative stress, reflecting the beta cells’ strained capacity to process and convert proinsulin to insulin efficiently to cope with increased demand for insulin (PMID: 36280617).

Building on this, we know that a range of inflammatory cytokines contribute to beta-cell stress in T2D, while IL-22 protects beta-cells from oxidative and ER stress regardless of the environmental triggers (PMID: 25362253 and 26576641). In animal models, exogenous IL-22 corrects much of the diabetes pathophysiology, however the complete restoration of glycaemic control is observed prior to any improvements in insulin sensitivity (PMID: 25362253). Similarly in this manuscript, we show that ablation of endogenous IL-22 signaling in the beta cells leads to ER stress, and increased proinsulin with age leading to hyperglycaemia. Although at 20 weeks of age, no changes in insulin resistance were observed, we did note that these mice develop insulin resistance with age (20 week old vs 52 week old insulin tolerance tests are shown here for the reviewer).

3. In lines 220 and 221, the authors claim, “Perhaps the endogenous protection by IL-22RA1 signaling could contribute to the lower rates of beta-cell dysfunction in females compared to males”. This statement appears to be misleading. The data shown in this manuscript indicates that the mice with beta cell specific IL-22ra1 deficiency develop glucose intolerance with age, which was more pronounced in female mice. This means that compared to male mice, female mice’s glucose tolerance is more dependent on IL22 signaling in beta cells. Therefore, this evidence cannot help explain why women have a lower incidence of T2D compared to men. A weakness of this manuscript is that the clinical relevance is unclear.

AR03: Thank you to the reviewer for this comment. We agree that the role of sex-differences in beta cell dysfunction is complex. We have now amended the statement to remove this speculation (Lines 235 - 237) as follows:

“Whilst there have been no reports of gender specific pancreatic islet IL-22RA1 expression in humans, to understand these sex-specific differences, further studies are needed to investigate whether sex hormones like estrogen can regulate the endogenous levels of IL-22ra1 or local IL-22 production in the islet.”

4. The authors have shown that IL22ra1 deficiency in pancreatic beta cells results in insufficient UPR, reduced islet growth and regeneration, and increased inflammatory cell infiltration. However, it is not clear which one is the main cause of glucose intolerance in beta cell specific IL22ra1 deficient mice.

AR04: Thank you to the reviewer for highlighting this point. We recognize the complexity in pinpointing a singular causative mechanism among insufficient unfolded protein response (UPR), reduced islet growth and regeneration, and increased inflammatory cell infiltration.

We have previously highlighted in a review that interleukin 22 protects β -cells from oxidative stress regardless of the environmental triggers and can correct much of diabetes pathophysiology in animal models (PMID: 26576641). Specifically, IL-22 has been shown to directly suppress UPR via the suppression of reactive oxygen species generation and protein misfolding in pancreatic beta cells. Moreover, others have demonstrated that IL-22 can induce regeneration of pancreatic beta cells (PMID: 25408874).

Our data in this paper suggests that these phenomena likely contribute in a synergistic manner to the glucose intolerance observed in beta cell-specific IL-22ra1 deficient mice. We hypothesize that IL-22RA1 signaling may act as a regulatory node that simultaneously influences these pathways, thereby maintaining beta-cell processes. We believe our findings contribute a foundational understanding of IL-22RA1's role in beta cell physiology and the complex interplay of mechanisms leading to glucose intolerance, setting the stage for more targeted investigations.

5. In lines 275-276, the authors explain that the ablation of IL-22ra1 may enhance beta-cell MHC II, leading to an increase in local islet inflammation. To support this view, the authors need to show increased MHC II in IL-22ra1 deficient beta cells.

AR05: Thank you to the reviewer for this suggestion, in light of this comment we have now conducted additional experiments to investigate whether there is a change in MHC II in the pancreatic beta cells in our model. Interestingly, our results demonstrate that there is an increase in MHC II on insulin positive cells in the IL-22ra1-deficient beta cells. Moreover, we treated islets with IFN γ to induce MHC II (as per previous publication: PMID 25959978) and demonstrate that IL-22 treatment can suppress MHC II induction on pancreatic beta cells. This data is now included in Fig. 5f and Supplementary Fig 10 demonstrating an increase in islet MHC-II is suppressed by IL-22.

Reviewer #2 (Remarks to the Author):

The manuscript entitled “Pancreatic Beta-Cell IL-22 Receptor Deficiency Induces Age-dependent Dysregulation of Insulin Biosynthesis and Systemic Glucose Homeostasis” by Sajiir et al., examined the effect of IL-22 Receptor ablation on glucose metabolism. The authors used a large variety of molecular, cellular, and morphological techniques to show that mice lacking the IL-22 receptor A1 have reduced insulin secretion, quality, and poor islet regeneration in addition to increased islet cellular stress and inflammation.

Comments

1. What does “high-quality” insulin mean?

AR06: In addressing the reviewer's question regarding the term "high-quality" insulin, we appreciate the opportunity for clarification. By "high-quality" insulin, we refer specifically to the biochemical and functional properties of insulin that are critical for its efficacy in glucose regulation. This distinction is particularly relevant in the context of our findings, where the ablation of IL-22ra1 in pancreatic beta cells was associated with an increased proinsulin: insulin ratio following glucose stimulated insulin secretion. This elevated ratio indicates a disruption in the normal processing and maturation of insulin, leading to a higher proportion of less mature and potentially less effective insulin precursor molecules, proinsulin. Furthermore, our observations of lowered glucose uptake by 3T3-L1 cells in response to this insulin underscores the functional implications of these biochemical alterations. These results collectively suggest that the "quality" of insulin, in terms of its proper folding, maturation, and functional capacity to facilitate glucose uptake, is compromised in the absence

of IL-22ra1 signaling. This compromised insulin quality directly impacts its effectiveness, highlighting the critical role of IL-22RA1 in maintaining insulin's functional integrity and, by extension, glucose homeostasis.

2: "ER" should be defined at first use

AR07: We thank the reviewer for highlighting this, we have now defined ER when it is first used.

3: The authors indicate that "... IL-22 treatment downregulated disease pathways...". What are these pathways?

AR08: We had used the Ingenuity Pathway Analysis (IPA) software to perform pathway analysis; the machine learning disease pathway function was used to determine key molecules that can affect diseases and their associated phenotypes. Through this, we identified that IL-22 treatment downregulated genes that were involved in endocrine pancreatic dysfunction, diabetes mellitus, severe pancreatic disorders, and impaired glucose tolerance. These were included in line 107 and we have altered text to clarify this as follows:

*"Importantly, IL-22 treatment downregulated **expression of key genes involved in disease pathways including involved-in** endocrine pancreatic dysfunction, Diabetes mellitus, severe pancreatic disorders, and impaired glucose tolerance (**Supplementary Fig. 2c**)."*

4: Lines 111-112: The author observed that "...animals developed increasingly severe glucose intolerance with age, which was more pronounced in female animals". The authors should explain this difference in more detail.

AR9: In response to the reviewer's request for additional detail regarding the observed sex-specific differences in glucose intolerance, we have updated the manuscript to include the following information (Lines 117 - 120: "Females *IL-22ra1^{ΔBKO}* animals exhibited onset of severe glucose intolerance beginning at 16 weeks of age (**Fig. 2c, d**). In contrast, male animals did not show a statistically significant change at this age. By 20 weeks, although not reaching statistical significance, there was a notable trend indicating that male animals were also on the trajectory towards glucose intolerance (**Supplementary Fig. 3a-b**)."

5: Line 169: "Antioxidant" is written in 2 formats. It would be nice to use the same format

AR10: We thank the reviewer for pointing this out, we have now made changes on line 105, used the format "antioxidant" throughout the manuscript.

6: Line 210: Please delete "(Fig 7)" from the discussion

AR11: We have now made this change as requested by the reviewer.

7: Line 364: Vendor details of the HFD should also be provided

AR12: We have now clarified the vendor details as "Speciality Feeds Australia, SF04-001" on line 397 and 398.

8: Line 410: Please replace "Histology" with "Immunofluorescence"

AR13: Thank you, this change has now been made to the manuscript.

9: Line 418: Which antibodies? At what dilution?

AR14: We have now incorporated supplementary table 4 (primary and secondary antibodies) into the methods text between lines 451 and 459.

10: Lines 478 & 509: Refs 2 & 17 are the same

AR15: Thank you to the reviewer for spotting this error – we have now changed updated the references.

Other Comments:

11: The authors should explain the possible role of IL-22RA on the plasma membrane of pancreatic alpha cells

AR16: We have now discussed the potential role of IL-22RA on pancreatic alpha cells (line 332).

“Pancreatic alpha cells play a key role in maintaining glucose homeostasis and are also known to express IL-22RA¹⁴⁴. However, in this study we show that reveal that alpha-cell-specific IL-22ra1 knockout (*IL-22ra1^{ΔαKO}*) does not impact key metabolic parameters in our mouse model. Specifically, these animals demonstrated no significant differences in body or pancreas weights, serum glucagon levels, glucose tolerance, insulin sensitivity, or endocrine/exocrine pathology when compared to littermate controls. Nonetheless, it is important to consider the potential roles of IL-22ra1 signaling in alpha cells that may not have been captured in the scope of our study. IL-22 is known for its role in tissue protection and regeneration, and whilst our data does not demonstrate a direct effect on the parameters measured, IL-22ra1 could potentially influence alpha cell stress responses, contribute to the maintenance of alpha cell mass, or play a role in intra-islet signaling that impacts the local islet microenvironment. Further research is necessary to elucidate these potential roles of IL-22RA1 in alpha cells, particularly under different physiological or pathological conditions that were beyond the scope of the current investigation.”

12: Why does the glucose level tail down to normal at 120 min after the glucose challenge in Figure 2c?

AR17: Thank you to the reviewer for this interesting question. Please also refer to AR02.

Intriguingly, in *IL-22ra1ΔβKO* animals, plasma glucose levels return to baseline post OGTT, suggesting that glucose normalization occurs independently of enhanced peripheral insulin sensitivity or response. Similarly in animal models, exogenous IL-22 administered glycaemic control even before improvements in insulin sensitivity.

This phenomenon suggests two potential and likely interconnected explanations. First, activation of insulin-independent glucose clearance mechanisms, possibly through upregulated activity of insulin-independent glucose transporters or alternative pathways that facilitate cellular glucose uptake, such as those stimulated by exercise or other physiological triggers. Second, this pattern could reflect the residual capacity of beta cells to eventually secrete enough insulin to manage glucose challenges, despite initial delays. It is important to note that *IL-22ra1^{ΔβKO}* mice eventually at 52 weeks of age develop insulin resistance, supporting a decline in this residual beta-cell function over time (data shown in AR02).

This text is now added to the discussion (line 252) to increase clarity for the readers.

REVIEWERS' COMMENTS

Reviewer #1 (Remarks to the Author):

The revised manuscript submitted by Sajiir et al. has satisfactorily addressed most of my previous concerns. However, it is a pity that due to insufficient research funding, the authors did not carry out single cell RNA-seq analysis of pancreatic islet cells. I agree with the authors that beta cell specific IL-22ra1 KO data somehow complement RNA-seq analysis of MIN6N8 cells and ddPCR analysis of dissected islets, but they cannot replace single cell RNA-seq analysis of pancreatic islet cells. Because of the absence of direct evidence of the beta cells' transcriptional profile, this revision still lacks sufficient scientific rigor.

Reviewer #2 (Remarks to the Author):

The authors have addressed all of the comments raised.

Reviewer #1 (Remarks to the Author):

Reviewer Comment: The revised manuscript submitted by Sajiir et al. has satisfactorily addressed most of my previous concerns. However, it is a pity that due to insufficient research funding, the authors did not carry out single cell RNA-seq analysis of pancreatic islet cells. I agree with the authors that beta cell specific IL-22ra1 KO data somehow complement RNA-seq analysis of MIN6N8 cells and ddPCR analysis of dissected islets, but they cannot replace single cell RNA-seq analysis of pancreatic islet cells. Because of the absence of direct evidence of the beta cells' transcriptional profile, this revision still lacks sufficient scientific rigor.

Author Response: We thank the reviewer for their understanding and for recognizing the efforts made to address the concerns previously raised. We acknowledge the reviewer's remarks regarding the potential value of single-cell RNA-seq analysis of pancreatic islet cells. Whilst we agree that such analysis could provide additional insights, due to the constraints of funding, we were unable to include it. Given the exhaustive revisions already undertaken, we have maintained the manuscript in its current form and hope that our work will encourage further research that might include such detailed analyses. We believe the current methodologies employed adequately support our conclusions and contribute valuable knowledge to the field.